# Household food sources and diarrhoea incidence in poor urban communities, Accra Ghana

Reuben Tete Larbi[1¤a], D. Yaw Atiglo[1], Maame B. Peterson[1¤b], Adriana A. E. Biney[1]*, Naa Dodua Dodoo[1], F. Nii-Amoo Dodoo[2]

1 Regional Institute for Population Studies (RIPS), University of Ghana, Legon, Accra, Ghana, 2 Department of Sociology and Criminology, The Pennsylvania State University, State College, Pennsylvania, United States of America

¤a Current address: Lancaster Environment Centre, Lancaster University, Lancaster, United Kingdom
¤b Current address: African Institute for Development Policy (AFIDEP), Lilongwe, Malawi
* abiney@ug.edu.gh

## Abstract

Diarrhoeal diseases remain a significant cause of morbidity and mortality, particularly in poor urban communities in the Global South. Studies on food access and safety have however not considered the sources of discrete food categories and their propensity to harbour and transmit diarrhoeal disease pathogens in poor urban settings. We sought to contribute to knowledge on urban food environment and enteric infections by interrogating the sources and categories of common foods and their tendency to transmit diarrhoea in low-income communities in Accra. We modelled the likelihood of diarrhoea transmission through specific food categories sourced from home or out of home after controlling for alternate transmission pathways and barriers. We used structured interviews where households that participated in the study were selected through a multi-stage systematic sampling approach. We utilized data on 506 households from 3 low-income settlements in Accra. These settlements have socio-economic characteristics mimicking typical low-income communities in the Global South. The results showed that the incidence of diarrhoea in a household is explained by type and source of food, source of drinking water, wealth and the presence of children below five years in the household. Rice-based staples which were consumed by 94.5% of respondents in the week preceding the survey had a higher likelihood of transmitting diarrhoeal diseases when consumed out of home than when eaten at home. Sources of hand-served dumpling-type foods categorized as "staple balls" had a nuanced relationship with incidence of diarrhoea. These findings reinforce the need for due diligence in addressing peculiar needs of people in vulnerable conditions of food environment in poor urban settlements in order to reap a co-benefit of reduced incidence of diarrhoea while striving to achieve the global development goal on ending hunger.

## Introduction

Diarrhoea is a major food-borne infectious disease that contributes significantly to the global disease burden. It is among the top five leading causes of death among children under five years, and accounts for about 2.5 million deaths each year globally [1]. Diarrhoea

**Data Availability Statement:** All relevant data are within the manuscript and its Supporting information files.

**Funding:** FNAD received a William and Flora
Hewlett Foundation grant which partly covered this
study. Other partial funders include International
Development Research Centre (IDRC) and
Research on Obesity and Diabetes among African
Migrants (RODAM). All funders had no role in
study design, data collection and analysis, decision
to publish, or preparation of the manuscript.

**Competing interests:** The authors have declared
that no competing interests exist.

disproportionately affects people in low-income communities who experience poor water
quality, poor sanitation and food contamination [2]. In low-income countries, children aged
three years and below experience on average of three episodes of diarrhoea annually [3]. The
2014 Ghana Demographic and Health Survey reports a 12% diarrhoea prevalence rate among
children below 5 years [4]. Amidst the rising chronic non-communicable disease prevalence,
infectious diseases remain major causes of morbidity in developing countries, putting much
pressure on health systems [5]. Africa has the highest diarrhoeal disease burden attributable
to exposure to contaminated food [6]. The transmission and spread of diarrhoeal disease
pathogens are mainly through contaminated food and water [7]. The 2015 World Health
Organisation (WHO) report on the burden of diseases attributable to the consumption of con-
taminated food reported 600 million episodes of illness, 420,000 deaths and 33 million disabil-
ity-adjusted life years (DALYs) [8]. Diarrhoeal epidemics have been associated with viral and
bacterial species such as norovirus, campylobacter, *Clostridium difficile*, and *Escherichia coli* [9,
10]. There is evidence that, in addition to stunting, recurrent diarrhoea episodes affect the cog-
nitive development of children [11]. This could also produce the ripple effect of increased risk
of mortality from other infectious diseases such as pneumonia, malaria and measles due to
reduced immunity [12, 13].

The growing proportion of global populations resident in informal settlements due to
population growth and rapid unplanned urbanisation presents several consequences, includ-
ing public health challenges resulting from poor water, sanitation and hygiene service provi-
sioning. Settlements in low-income countries tend to have a unique food environment that is
characterised by a wide array of street foods prepared and served under unhygienic conditions.
There is therefore a higher risk of faecal contamination to food and drinking water.

Several studies have established a relationship between sanitation, water quality, and diar-
rhoea incidence [2, 14]. Other studies have examined the levels of contamination or the micro-
bial loads in some categories of street foods [14–16]. Whilst food sampled from hotels in Accra
had acceptable levels of microbial contamination, street foods mainly from food vendors had
higher loads of pathogens [15]. Similarly, there are observed differential levels of microbial con-
tamination by food types. For instance, lower microbial loads were found in kenkey (boiled
corn dough wrapped in corn leaves) and waakye (a mixture of boiled rice and beans), but higher
loads of *Staphylococcus aureus* and *E. coli* were found in fufu (boiled cassava and plantain
pounded in a mortar and mixed with the hands) [17]. In addition, most ready-to-eat (RTE)
foods sampled from Kumasi, a major city in Ghana, had levels of microbial loads that were
above the acceptable limits [18]. Infant feeding practices or contamination of weaning foods and
their effects on infant diarrhoea have also been documented [19, 20]. There are however knowl-
edge gaps in the relationship between the sources of different types of tropical household foods
and diarrhoea incidence. It is common knowledge that food is a major conduit in the diarrhoeal
disease transmission pathway, but in order to enhance educational and regulatory campaigns on
faecal-contaminated foods, the different categories of common foods in low-income urban set-
tings and their propensity to transmit diarrhoea need more thorough interrogation. Thus, the
novelty of this study is the classification of common tropical foods by source and frequency of
consumption vis-a-vis the likelihood of transmitting diarrhoeal disease. We seek to identify the
sources and categories of food that have the likelihood to transmit diarrhoeal disease pathogens.

## Food sources and diarrhoeal infection in Ghana

Microbial contamination of food can occur at any stage from where food is produced, trans-
ported and processed, to where it is prepared, sold and consumed [21]. Hence, hygienic prepa-
ration and storage of food, both at home and by vendors, is particularly important in reducing

the risk of contamination. Apart from food prepared at home, there are several out of home food sources including restaurants, "chop bars" (an informal eatery where local staples are sold), and street vendors that are widely patronised in Ghana, just as in other countries in the Global South. Street foods are often used to refer to an array of RTE foods sold and often prepared in public places [16]. These foods are either eaten where they are sold in public spaces or taken away in plastic bags, polystyrene packs or leaves. Street foods are therefore major sources of oro-faecal pathogen transmission, as there is a plethora of potential sources of contamination. These sources of contamination include the water used, cutlery or equipment, hands, mode of preparation or handling, ingredients, bowls or storage containers [22]. About 77% of food-borne diseases are due to preparation and handling practices at food establishments [23]. We therefore hypothesised a higher likelihood of diarrhoea incidence in households that more frequently consume out-of-home staples. We defined a household as "a person or a group of persons, who live together in the same house or compound and shared the same house-keeping arrangements and recognise one person as the head" [24].

## Food environment in low-income urban communities

RTE foods have increasingly become central to the food supply chain of the urban population in both developed and developing countries [25]. This is due to the availability and affordability of these foods compared to the time and financial cost of cooking at home [22, 26]. Thus, RTE foods offer easy access to traditional staples that are arduous and time consuming to prepare at home [18, 27]. These foods are often prepared and/or sold at school compounds, lorry stations, and along busy roads [27, 28]. Another emerging trend in the urban food environment is the proliferation of foreign-style fast foods which are now easily accessible to persons in the different socio-economic milieus at different geographical locations and food establishments. These include a similitude of the Asian *nasi goreng* which is locally called fried rice, or a mix of noodles and vegetables popularly called "Indomie" (after the Indomie brand), and salads [27]. There is also an abundance of cut fruits that are packed in polythene bags or transparent plastic containers sold on the streets. It is easy to purchase these fruits from street vendors while stuck in vehicular traffic. Traditional dumpling-type staples including banku, kenkey, fufu, rice balls etc., herein referred to as staple balls, often served with soups and stews, are sold at designated enclosed places called "chop bars". These staples are also available at restaurants and hotels at higher prices. Another source of street food is the "mobile vendors" who carry foods on their heads or on hand-drawn carts and serve customers in plates or wrapped in large leaves. Where plates are used, the vendor cleans the plate and spoons with a piece of sponge immersed in water before serving the next customer. A modernised form of the "mobile vendors" is the food vans where food is served in disposable plastic plates or containers.

Hygiene practices among food vendors tend to be below recommended standards [29, 30]. Thus, the levels of total microbial load in their foods have been found to exceed the acceptable limit of $10^5$ Colony Forming Units per gram (cfu/g) set by the Ghana Standards Authority [15, 18, 30]. In addition, some street foods are more prone to microbial contamination than others due to the nature of the preparation and serving procedures and the contamination pathways that these processes create [26]. Further, some studies have reported a generally low level of education among food vendors in major cities in Ghana [18, 31].

The decision to buy food from a particular vendor is often contingent on price, consumer and vendor relationships, appearance of environment, appearance of the vendor, taste and accessibility of the food. Consumers often prioritize affordability and accessibility over food safety. Similarly, vendors are often more concerned about their own appearance while vending than basic food hygiene and safety practices while preparing food [29]. Ultimately, consumers

have more control over the content and quality of what they eat if they prepare their own food at home.

### Analytical framework: The F-diagram of faecal-oral transmission route

Most diarrhoeal diseases are caused by pathogens found in human excreta which are transmitted to the mouth through a complex web of pathways. Food and drinking water are significant routes in the oro-faecal transmission process, though pathogens could be passed directly from intermediary "Fs"- fingers, fluids which refers to drinking water in this study, flies, and field/ floors to the new host (Fig 1). Children could get infected by introducing contaminated objects including fingers into their mouth directly and not necessarily through food. Safely contained faecal matter is therefore a primary barrier in the transmission route. Water, sanitation and hygienic (WASH) practices which include regular handwashing with soap at critical times such as before eating, and treatment of drinking water form a secondary barrier. These could curtail the transmission of pathogens from contaminated floors, fingers, flies (and other insects such as cockroaches), and fluids such as water to the individual. The F-diagram shows the significant role of food in the faecal-oral transmission process [32]. Often, foods prepared and eaten at public places are more exposed to contamination through the intermediary "Fs" than those prepared and eaten at home. However, this pathway has not been tested in infectious disease transmission models in urban poor settings characterised by poor sanitation and hygiene conditions within and outside homes. In this paper, we test the transmission pathway through food types and place of preparation and consumption after controlling for alternate transmission pathways.

## Materials and methods

### Study area

The study was conducted in three densely populated poor urban communities in Accra: James Town, Ussher Town and Agbogbloshie. These were purposely selected based on their indigeneity, geographical location and a migrant settlement that has less threat of eviction, so that the research team would be able to compare the characteristics and effects of urban poverty

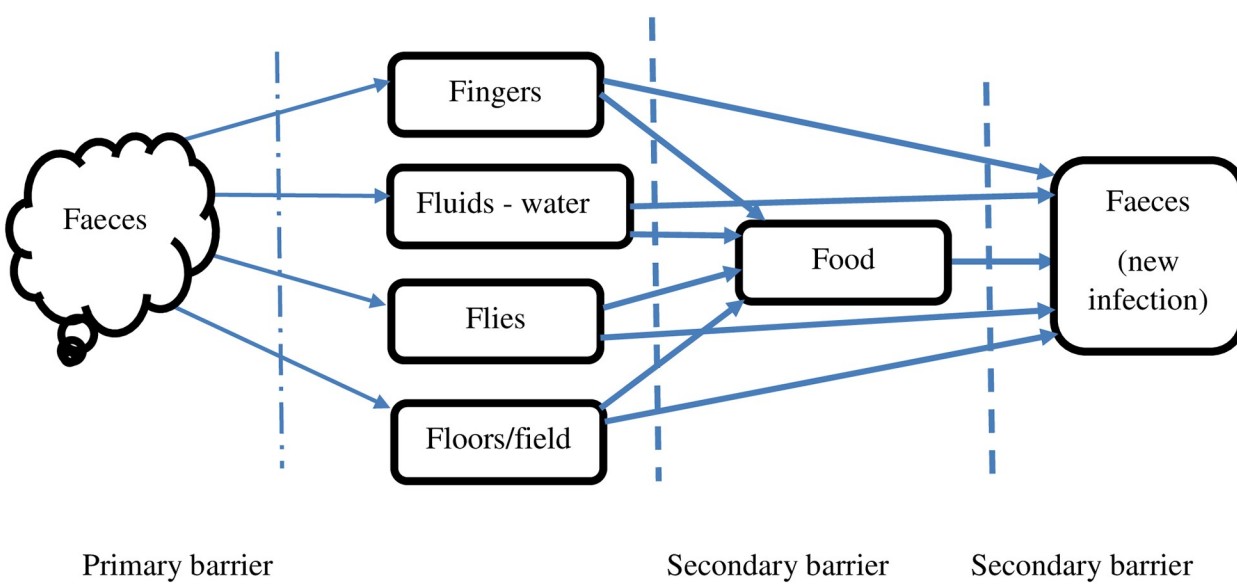

**Fig 1. F-diagram of faecal-oral disease transmission pathways.** Source: Wagner, Lanoix & WHO (1958).

between communities of different migratory and natal circumstances. The communities are known for extensive street food vending [33]. James Town and Ussher Town are adjacent to each other and constitute the Ga-Mashie Traditional Area. They are inner city areas that have been outpaced by socio-economic development. Agbogbloshie, on the other hand, is a typical migrant slum and market area, though some residents trace their roots there to as far back as the 1960s [34]. Residents of James Town exhibit relatively higher socio-economic characteristics than the other two localities. The communities are known for poor sanitation conditions, non-existent or limited access to toilet facilities, and lack of potable water in some parts. Waste management is a major challenge as drains are often open and choked with garbage [35]. Toilet facilities are often owned communally or by a private individual and used by several households. Commercialisation of toilets and baths is common in these localities, particularly in Agbogbloshie [35–38]. The combination of poor sanitation and inadequate drainage systems make some areas of the study sites flood-prone, which poses an additional threat to residents' health [35]. Life in these three settlements typifies experiences in informal settlements in sub-Saharan Africa [39].

## Data collection—Sampling, fieldwork and ethics

This paper used data from the third wave of the Urban Health and Poverty Project survey by the Regional Institute for Population Studies (RIPS) at the University of Ghana. The survey was conducted between September and October 2013 in three communities—James Town, Ussher Town and Agbogbloshie using the enumeration areas (EAs) demarcated by the Ghana Statistical Service. Twenty-nine EAs were randomly sampled using the random number generator; eight in James Town, sixteen in Ussher Town and five in Agbogbloshie. The number of EAs selected in each locality was proportionate to the population size of that locality. All households in the EAs were listed, after which a total of forty households were systematically sampled from each EA. The total number of household and individual interviews conducted was 660 and 782, respectively. For this paper, data on 506 households were used; households that did not have individuals responding to the food consumption schedule were excluded. Data collection was conducted by four teams. Each comprised a supervisor, a questionnaire editor (who ensured that discrepancies in responses were corrected), a mediator (who assisted the supervisor to ensure that the right households are interviewed), ten enumerators conducting interviews, and two others who took anthropometric measurements such as blood pressure, weight and height measurements of respondents. All interviews were reviewed by the questionnaire editor and approved by the team supervisor before double entry into the database.

We used two sets of questionnaires—a household questionnaire and an individual questionnaire for collecting the data. The household questionnaires were administered to de facto household heads. After this, individual questionnaires were administered to eligible household members (men aged 15 to 59 years and women aged 15 to 49 years). The interviews were conducted in either English, or one of the dominant local languages spoken in this area; Ga and Twi. Prior to data collection, ethical approval was obtained from the Noguchi Memorial Institute for Medical Research (NMIMR) Institutional Review Board at the University of Ghana (certified Protocol Number 105/12-13). Respondents consented to participate in the study after the purpose of the study and their rights to participate were explained to them by appending their signatures, or a thumbprint in cases where they could not write, on informed consent forms.

## Variables

**Outcome variable.** In this study, diarrhoea incidence was measured as the last time a member of the household had diarrhoea. Responses ranged from "less than a week ago" to

"more than a month ago". As a check on the consistency of the response, a follow up question enquired about the number of times a household member had diarrhoea in the past 12 months. This was subsequently coded into a dichotomous measure on whether the household recorded any cases of diarrhoea in the past 12 months.

**Main independent variable.** Food categories and sources of consumption were measured through household heads' reports about the number of times they consumed specified staple foods in the seven days preceding the survey. Respondents indicated the number of times they ate specific foods from a list of 35 food items listed under twelve categories that represent common foods consumed in the study communities. Respondents also indicated where they consumed those staples, whether in homes, restaurants, chop bars or from street vendors. An index of the proportion of each staple food group consumed at home as compared to out-of-home was computed. This was done because many food items were mainly consumed at home or from a street vendor. Foods were broadly categorised under "staple balls", "staple rice" and fruits. Staple balls are named so because they are served as balls and include foods such as fufu, banku, kenkey, rice balls etc. Staple rice refers to rice-based dishes that are served and eaten as grains including boiled rice, waakye (a mix of rice and beans), jollof/fried rice etc. Definitions of these foods in terms of preparation, handling and service are detailed elsewhere [16]. The equations below indicate the proportion of home to out-of-home food source calculations.

Sum of all staples taken at source $i$

$$Source\ of\ food = \sum\nolimits_{w=1}^{n} Staples(i) \tag{1}$$

$w$ = staple rice, staple balls or fruits.

$i$ = individual food items under each staple category ["Staple rice" comprised waakye, fried rice, jollof rice and plain rice; "staple balls" comprised fufu, banku, akple, tuo zaafi, kenkey, rice-balls; "fruits" comprised oranges, mangos, pineapple, pawpaw/papaya, watermelon, apples, grapes, avocado pears and bananas].

The sum of a particular staple from all sources was computed as

$$Source\ sum = \sum_{j=1}^{n} Staples\ (j) \tag{2}$$

$j$ = home, restaurants, chop bars or street vendor.

The proportion of staples taken at home was computed as

$$source\ (k) = \frac{home}{\sum staples\ (j)} * 100\% \tag{3}$$

This generated a source of food variable (k) ranging from 0 to 100 which was finally coded as

$$
\begin{aligned}
k &= 0 & &Away\ only \\
0.1 &< k < 99.9 & &Both\ home\ and\ away \\
k &= 100 & &Home\ only \\
k &= sys\ missing & &Did\ not\ take\ this\ staple
\end{aligned}
$$

Households with a score of 0 ate the staple food solely outside the home; those with a score of 100 ate the staple exclusively at home, and the rest of the households between these extremes ate in both settings to varying degrees.

In grouping foods, we assumed that soups and stews are eaten with staple rice and staple balls and so the former were not grouped as distinct food items. Also, we assumed that food consumption behaviour in the week preceding the survey represents the household consumption pattern.

**Control variables.** The control variables selected for inclusion were source of drinking water, washing of hands at specific critical times, household size, wealth category, number of children under 5 years, type of toilet facility, solid/liquid waste disposal, and educational level and sex of household head. These variables contribute to diarrhoea prevalence by enhancing or inhibiting transmission [40–42]. Household wealth was computed using assets owned by the household such as a car, mobile phone, radio, television and fishing boat. We also included the materials used to construct the dwelling in the measurement of wealth. A household wealth index was created using the first factor generated from a principal component analysis. The index was then grouped into terciles; the lowest third in the index were labelled as poor households, the moderate households were the middle third, with the top third in the index being the rich.

## Statistical analysis

We first explored the frequency distributions of the outcome and main independent variables, as well as socio-demographic and behavioural characteristics of respondents. We then checked bivariate associations between the outcome variable and independent variables using chi square tests. Variables including sex of household head, educational attainment of household head, study locality, toilet facility, household wealth and handwashing with soap before eating are all essential variables that are theoretically linked to diarrhoeal incidence in a household [36, 45]. Hence, though there was no statistical association between these variables and diarrhoea incidence at the bivariate level, we still included them in the model. The authors ran variance inflation factor (VIF) tests to assess multicollinearity between explanatory variables before including them in the regression model. Each variable was compared with the others while monitoring the collinearity statistics. Conclusions were made from the binary logistic regression model which tested the amount of variance in household diarrhoea incidence that is significantly explained by food sources only on one hand, and then after controlling for all exposure and behavioural factors that limit or promote diarrhoeal disease transmission.

## Results

The univariate analysis indicates 506 households were included in the study. As shown in Table 1, out of the 506 households, more than half (54.7%) reported at least one diarrhoea episode in the 12 months preceding the survey. With regards to food types and place of consumption, within the week preceding the survey about two-fifths of households consumed staple rice or staple balls both at home and outside, about a third ate staple rice or staple balls out of home and about a fifth ate rice-based meals or staple balls only prepared at home. Small proportions did not consume staple balls or staple rice during the week preceding the survey. Fruits, on the other hand, were mainly eaten outside the home by a significant proportion of households, and less than a tenth consumed fruits served at home (8.5%). About one in every five respondents did not eat any fruit in the week preceding the survey (Table 1).

Other variables that have been shown to influence the transmission of diarrhoea are the type of toilet facility, solid waste disposal, liquid waste disposal, and source of household water. About three-quarters of households (78.4%) used public toilets, a little below a tenth had a private KVIP (9.1%) and private water closet (9.3%). The remaining households used pan bucket latrines or had no toilet facilities and so used the bush or beach. A little more than half of

**Table 1. Descriptive statistics of diarrhoea incidence, food types and sources and exposure variables.**

| Variables | Number | Percent |
|---|---|---|
| **Diarrhoea incidence** | | |
| No | 229 | 45.3 |
| Yes | 277 | 54.7 |
| **Food types and sources** | | |
| **Staple rice** | | |
| Ate outside only | 176 | 34.8 |
| Ate both home and outside | 210 | 41.5 |
| Ate home only | 92 | 18.2 |
| Did not eat staple rice | 28 | 5.5 |
| **Staple balls** | | |
| Ate outside only | 174 | 34.4 |
| Ate both home and outside | 211 | 41.7 |
| Ate home only | 107 | 21.1 |
| Did not eat staple balls | 14 | 2.8 |
| **Fruits** | | |
| Ate outside only | 318 | 62.8 |
| Ate both home and outside | 36 | 7.1 |
| Ate home only | 43 | 8.5 |
| Did not eat fruits | 109 | 21.6 |
| **Exposure variables** | | |
| **Toilet facility** | | |
| No facility/bucket/pan | 16 | 3.2 |
| Home WC | 47 | 9.3 |
| Home KVIP | 46 | 9.1 |
| Public | 396 | 78.4 |
| **Household water source** | | |
| Indoor plumbing | 101 | 20.0 |
| Private outside | 277 | 55.0 |
| Public outside | 126 | 25.0 |
| **Liquid waste disposal** | | |
| Septic tank | 9 | 1.8 |
| Community drain | 430 | 85.0 |
| Indiscriminately | 67 | 13.2 |
| **Solid waste disposal** | | |
| Collected by company | 234 | 46.3 |
| Waste container | 74 | 14.7 |
| Cart pushers | 170 | 33.7 |
| Other | 27 | 5.3 |
| **Educational level of head** | | |
| No education/preschool | 72 | 14.2 |
| Primary | 97 | 19.2 |
| JHS/Middle | 205 | 40.5 |
| Higher | 132 | 26.1 |
| **Locality name** | | |
| Agbogbloshie | 81 | 16.0 |
| James Town | 138 | 27.3 |
| Ussher Town | 287 | 56.7 |

(*Continued*)

**Table 1.** (Continued)

| Variables | Number | Percent |
|---|---|---|
| **Presence of children under 5** | | |
| No | 368 | 72.7 |
| Yes | 138 | 27.3 |
| **Sex of household head** | | |
| Male | 261 | 51.6 |
| Female | 245 | 48.4 |
| **Household size** | | |
| 1 | 103 | 20.4 |
| 2–3 | 148 | 29.2 |
| 4–5 | 147 | 29.1 |
| 6+ | 108 | 21.3 |
| **Household wealth** | | |
| Poor | 210 | 41.5 |
| Moderate | 107 | 21.1 |
| Rich | 189 | 37.4 |
| **Washed hands with soap and water before preparing food** | | |
| Yes | 138 | 27.3 |
| No | 368 | 72.7 |
| **Washed hands with soap and water before eating** | | |
| Yes | 411 | 81.2 |
| No | 95 | 18.8 |
| **Total** | **506** | **100.0** |

respondents (55%) used private water located outside the building, a quarter used a public water source located outside (25%), and the rest had indoor plumbing (20%). Also, whilst the majority of households (85.0%) disposed their liquid waste into a community drain, a little over a tenth (13.2%) disposed of it indiscriminately, and the rest had septic tanks. With regards to solid waste, it was collected by a company in over two-fifths (46.3%) of households. One in every seven households disposed of solid waste at designated waste container sites, about a third (33.7%) paid for the services of ad hoc cart operators, and the rest reported other forms of disposal such as burning or burying. Less than a third of households (27.3%) had children under 5 years of age. Also, female household headship (48.8%) was higher than the 2010 national urban average of 38% [43]. A large proportion of respondents (73.5%) did not wash their hands with soap and water before preparing food on the day preceding the survey. However, about eight in every ten respondents washed their hands with soap and water before eating.

The association between the food sources, other factors and the incidence of diarrhoea was tested in the next stage of the analysis. The source of staple rice was significantly associated with the household diarrhoea incidence, while sources of fruits and staple balls were not. Also, the disposal of household liquid and solid waste, household size, water source, and the presence of children under five years were significantly associated with household diarrhoea incidence. As Table 2 shows, the proportion of households that reported diarrhoea was high among those who consumed staple rice "outside home only" or "both at home and outside home" (54.7% and 58.1% respectively). In addition, households where liquid waste was disposed indiscriminately reported higher diarrhoea incidence than households that disposed of liquid waste into septic tanks or community drains. Contrary to our expectation, households

**Table 2. Bivariate associations between the incidence of diarrhoea and food sources, and exposure factors.**

| Variables | Diarrhoea incidence | | |
|---|---|---|---|
| | Reported diarrhoea | Total respondents | |
| | Number (%) | Number | $\chi^2$ |
| **Staple balls** | | | **3.43** |
| Ate outside only | 87 (50.0) | 174 | |
| Both home and outside | 125 (59.2) | 211 | |
| Home only | 57 (53.3) | 107 | |
| Did not eat | 8 (57.1) | 14 | |
| **Staple rice** | | | **7.318*** |
| Ate outside only | 101 (57.4) | 176 | |
| Both home and outside | 122 (58.1) | 210 | |
| Home only | 44 (47.8) | 92 | |
| Did not eat | 10 (35.7) | 28 | |
| **Fruits** | | | **2.657** |
| Ate outside only | 166 (52.2) | 318 | |
| Both home and outside | 20 (55.6) | 36 | |
| Home only | 27 (62.8) | 43 | |
| Did not eat | 64 (58.7) | 109 | |
| **Sex of household head** | | | **0.481** |
| Male | 139 (53.3) | 261 | |
| Female | 138 (56.3) | 245 | |
| **Education of head** | | | **2.592** |
| No education/preschool | 45 (62.5) | 72 | |
| Primary | 55 (56.7) | 97 | |
| JHS/Middle | 107 (52.2) | 205 | |
| Higher | 70 (53.0) | 132 | |
| **Locality name** | | | **2.573** |
| Agbogbloshie | 45 (55.6) | 81 | |
| James Town | 83 (60.1) | 138 | |
| Ussher Town | 149 (51.9) | 287 | |
| **Liquid waste disposal** | | | **18.575*** * |
| Septic tank | 5 (55.6) | 9 | |
| Community drain | 219 (50.9) | 430 | |
| Indiscriminately | 53 (79.1) | 67 | |
| **Solid waste disposal** | | | **11.320*** |
| Collected by company (R) | 130 (55.6) | 234 | |
| Refuse container | 52 (70.3) | 74 | |
| Cart pushers | 82 (48.2) | 170 | |
| Other | 12 (44.4) | 27 | |
| **Presence of children under 5** | | | **3.664*** * |
| No | 211 (57.3) | 368 | |
| Yes | 66 (47.8) | 138 | |
| **Toilet facility** | | | **6.055** |
| No facility/bucket/pan | 13 (81.2) | 16 | |
| Home WC | 22 (46.8) | 47 | |
| Home KVIP | 27 (58.7) | 46 | |
| Public | 215 (54.3) | 396 | |
| **Household water source** | | | **9.638*** * |

*(Continued)*

**Table 2.** (Continued)

| Variables | Diarrhoea incidence | | |
|---|---|---|---|
| | **Reported diarrhoea** | **Total respondents** | |
| | **Number (%)** | **Number** | **$\chi^2$** |
| Indoor plumbing | 52 (51.5) | 101 | |
| Private outside | 140 (50.5) | 277 | |
| Public outside | 84 (66.7) | 126 | |
| **Household size** | | | **9.285**\*\* |
| 1 | 61(59.2) | 103 | |
| 2–3 | 67 (45.3) | 148 | |
| 4–5 | 91 (61.9) | 147 | |
| 6+ | 58 (53.7) | 108 | |
| **Household wealth** | | | **4.153** |
| Poor | 126 (60.0) | 210 | |
| Moderate | 53 (49.5) | 107 | |
| Rich | 98 (51.9) | 189 | |
| **Wash hands with soap before preparing food** | | | **3.096**\* |
| Yes | 30 (44.8) | 67 | |
| No | 242 (56.3) | 430 | |
| **Wash hands with soap before eating** | | | **.467** |
| Yes | 115 (53.0) | 217 | |
| No | 157 (56.1) | 280 | |
| **Total** | 277 (54.7) | 506 | |

\*p < .05;

\*\*p < .01;

\*\*\* p < .001.

with children below five years reported less diarrhoea incidence than those that did not have children. However, the number of diarrhoea cases per household increased with household size. Further, households that used public water located outside their compounds reported higher diarrhoea incidence than those who used other water sources. Finally, households that reported hand washing with soap and water before eating had fewer cases of diarrhoea than those who did not.

The final stage of the analyses was to examine the proportion of variance in household diarrhoea incidence that is explained by food sources and other exposure and limiting factors. The results of the stepwise binary logistic regression models are presented in Table 3. We have reported the adjusted odds ratios (AOR) for both models. Model 1 examines the independent relationship between household food sources and diarrhoeal disease incidence. The model chi square test (17.24, p < .05) indicates that the binary logistic regression model fits the data. Secondly, the model indicates that household food sources independently explain 4.2 percent of the variation in diarrhoeal disease incidence among households. The highest VIF value from the collinearity statistics was 1.39 which falls below the acceptable threshold value of 5. Hence, no variables were collinear and could be used in the regression model. Of the three different food categories, the results indicate a significant relationship between staple rice and diarrhoea incidence. Households that consumed more rice-based foods such as waakye, jollof rice, plain rice and fried rice at home, have lower likelihood of contracting diarrhoeal disease that those who ate it outside the home.

**Table 3. Binary logistic regression models displaying the relationship between diarrhoeal incidence and food types and source, and other exposure factors.**

| Variables | Model 1 | | Model 2 | |
|---|---|---|---|---|
| | AOR (std. error) | 95% CI | AOR (std. error) | 95% CI for AOR |
| **Food types and sources** | | | | |
| **Staple rice** | | | | |
| Ate outside only (RC) | 1.00 | | 1.000 | |
| Both home and outside | .858 (.225) | [.552, 1.334] | .908 (.253) | [.553, 1.491] |
| Ate home only | .517 (.292)** | [.292, .917] | .515 (.325)** | [.272, .973] |
| Did not eat this staple | .328 (.445)* | [.137, .784] | .311 (.479)** | [.122, .795] |
| **Staple balls** | | | | |
| Ate outside only (RC) | 1.00 | | 1.000 | |
| Both home and outside | 1.676 (.225) | [1.077, 2.606] | 1.671 (.257)** | [1.009, 2.766] |
| Ate home only | 1.351 (.280) | [.781, 2.336] | 1.589 (.317) | [.853, 2.960] |
| Did not eat this staple | 1.681 (.601) | [.518, 5.458] | 2.470 (.642) | [.701, 8.695] |
| **Fruits** | | | | |
| Ate outside only (R) | 1.00 | | 1.000 | |
| Both home and outside | 1.220 (.363) | [.599, 2.484] | 1.073 (.403) | [.487, 2.365] |
| Ate home only | 1.847 (.353) | [.925, 3.687] | 1.879 (.403) | [.853, 4.139] |
| Did not eat fruits | 1.484 (.235) | [.937, 2.351] | 1.241 (.265) | [.739, 2.086] |
| **Exposure variables** | | | | |
| **Toilet facility** | | | | |
| No facility/bucket/pan (R) | | | 1.000 | |
| Home WC | | | .390 (.777) | [.085, 1.788] |
| Home KVIP | | | .408 (.790) | [.087, 1.919] |
| Public | | | .429 (.716) | [.106, 1.746] |
| **Household water source** | | | | |
| Indoor plumbing (R) | | | 1.00 | |
| Private outside | | | 1.056 (.323) | [.629, 1.772] |
| Public outside | | | 1.985 (.257)** | 1.072, 3.677] |
| **Liquid waste disposal** | | | | |
| Septic tank | | | 1.000 | |
| Community drain | | | .479 (.769) | [.106, 2.166] |
| Indiscriminately | | | 1.555 (.829) | [.306, 7.904] |
| **Solid waste disposal** | | | | |
| Collected by company (R) | | | 1.000 | |
| Refuse container | | | 1.056 (.299) | [.588, 1.895] |
| Cart pushers | | | .654 (.230)* | [.416, 1.027] |
| Other | | | 1.423 (.503) | [.531, 3.816] |
| **Educational level** | | | | |
| No education /preschool(R) | | | 1.000 | |
| Primary | | | 1.102 (.371) | [.533, 2.279] |
| JHS/Middle | | | 1.082 (.331) | [.566, 2.070] |
| Higher | | | .843 (.261) | [.506, 1.406] |
| **Locality name** | | | | |
| Agbogbloshie (R) | | | 1.000 | |
| James Town | | | 1.230 (.345) | [.626, 2.417] |
| Ussher Town | | | .967 (.311) | [.526, 1.780] |
| **Presence of children under 5** | | | | |
| No (R) | | | 1.000 | |

**Table 3.** (Continued)

| Variables | Model 1 | | Model 2 | |
|---|---|---|---|---|
| | AOR (std. error) | 95% CI | AOR (std. error) | 95% CI for AOR |
| Yes | | | .544 (.258)** | [.328, .902] |
| **Sex of household head** | | | | |
| Male (R) | | | 1.000 | |
| Female | | | 1.221 (.245) | [.756, 1.973] |
| **Household size** | | | | |
| 1 (R) | | | 1.000 | |
| 2–3 | | | .541 (.322) | [.288, 1.017] |
| 4–5 | | | 1.471 (.347)* | [.745, 2.905] |
| 6+ | | | 1.210 (.377) | [.578, 2.535] |
| **Household wealth** | | | | |
| Poor (R) | | | 1.000 | |
| Moderate | | | .631 (.278)* | [.366, 1.088] |
| Rich | | | .705 (.252) | [.430, 1.154] |
| **Wash hands with soup and water before preparing food** | | | | |
| Yes (R) | | | 1.000 | |
| No | | | 1.436 (0233) | [.909, 2.269] |
| **Wash hands with soap and water before eating** | | | | |
| Yes (R) | | | 1.000 | |
| No | | | .749 (.262) | [.448, 1.252] |

*p < .05;

**p < .01;

*** p < .001 N = 506

($\chi 2 = 17.24$, p<0.05) Nagelkerke $R^2 = 0.045$; ($\chi 2 = 89.53$, p<0.001) Nagelkerke $R^2 = 21.8$

After controlling for other background factors in Model 2, a total of 21.8 percent of variance in household diarrhoea incidence was explained by the independent variables in the model. The model reveals that households that consumed staple rice from home only and those who did not eat any rice staple were less likely to report diarrhoea than those who consumed staple rice from out of home. Secondly, households that ate staple balls from both home and outside had a higher likelihood of reporting diarrhoea than those who ate stable balls outside the home only. Households that ate staple balls only at home had higher odds ratio of reporting diarrhoea but were not significantly different from those who consumed from only outside the home. There was no significant relationship between eating fruits from any source and diarrhoea incidence. With regards to other diarrhoea exposing factors, households that used public water facilities outside the home were more likely to report diarrhoea than those who had indoor plumbing. In addition, households with one or more children below five years were less likely to report diarrhoea than households without a child aged below five years. Washing of hands with soap and water did not directly explain the variance in diarrhoeal disease incidence. However, the likelihood of households that ate staple balls both at home and outside the home reporting diarrhoea declined after controlling for washing of hands with soap and water. Finally, at a 90% confidence level, households with moderate wealth had a lower likelihood of reporting diarrhoea than poor households. There was also a higher likelihood of diarrhoeal disease incidence in households with 4–5 members compared to single member households, as expected.

## Discussion

We examined the relationship between sources of different types of household foods and diarrhoeal disease incidence. We hypothesized that households that consume different categories of out-of-home foods would have higher diarrhoeal disease incidence than those who consume them at home. Following the F-diagram, we modelled the faecal-oral diarrhoeal disease transmission pathways through food contamination. The source of staple rice significantly determines household diarrhoea incidence. In low-income urban settlements in Accra, households that mainly eat rice-based foods at home have a lower likelihood of reporting diarrhoea compared to those who eat them out of home. This observation corroborates our hypothesis and extant literature, which indicate higher levels of microbial loads in street foods [29]. From the data used in this study, staple rice eaten out of home is mainly purchased from street vendors where risk of microbial contamination is high. Food can become contaminated with *Vibrio cholerae*, for instance, from infected, convalescent and even asymptomatic food handlers. Contamination mainly originates from food preparation and handling, including the water used in preparation, the source of vegetables or ingredients and plates and cutlery used for serving. In home environments, there may be less of these contaminants. Also, out-of-home rice-based dishes are usually served with macaroni and salads which have the highest potential for transmitting diarrhoeal pathogens [16]. Foods such as rice, vegetables, sea food, potatoes, and cucumbers have been found to harbour diarrhoeal disease-causing pathogens [44]. Inadequate washing and cooking of these foods, which often applies to food vendors, facilitates transmission of these pathogens to consumers. Secondly, households that eat staple balls both at home and out of home have a higher likelihood of reporting diarrhoea compared to those who eat staple balls outside only. The same observation is made at the bivariate level of the analysis where higher proportions of households that eat staple balls both at home and outside reported diarrhoea, though the proportions are not significantly different. This finding is unexpected but could suggest that those who eat both home and outside sources of staple balls are prone to microbial contamination, which may be attributed to the mode of preparing and serving staple balls. For example, fufu (a major traditional staple in Ghana) is pounded in the mortar and mixed with the hands, often under low temperature conditions. It is dished into an earthenware bowl and often served with soup. The food is then eaten with the hands and often gets cold before the meal is finished. Banku is another common staple ball which is eaten with the hands and is commonly eaten at home and from street vendors. Thus, staple balls are prone to contamination if personal hygiene such as washing of hands during preparation and before eating is not maintained. Kenkey on the other hand is boiled in the husk of maize after mixing and moulding with the hands; serving and eating however, presents a risk of contamination as both are done with the hands. It is not typically made at home but mainly purchased from street vendors. Further, soups that are often eaten with stable ball meals are prone to contamination from a myriad of sources including water used in cooking and mode of food storage. Ideal temperature conditions could exacerbate the proliferation of pathogens caused by unsafe food storage practices. For instance, non-01 *V. cholerae* is able to grow between 20˚C and 45˚C and within a wide range of pH in different media [44]. Contaminated soups and staple balls could therefore harbour such pathogens for protracted periods. As depicted in the F-diagram on secondary barriers to faecal-oral disease transmission, the likelihood of people who consume staple balls reporting diarrhoea abates when washing of hands is controlled for. Wasihun et al. [45] found washing of hands at critical times as one of the predictors of diarrhoea in a study conducted in Ethiopia. This is evidenced at the bivariate analysis where a significantly higher proportion of households that do not wash hands with soap and water before preparing food reported diarrhoea. However, from our model, we did not find handwashing as a significant predictor of household diarrhoea incidence.

In addition to the source of food, source of household water significantly explains the variance in diarrhoea incidence. Households that use public water located outside the home have a higher likelihood of reporting diarrhoea than those who have indoor plumbing, which is consistent with existing knowledge [2, 46, 47]. Transporting water from source to point of consumption and storage of water in contaminated environments tends to compromise water quality [48]. On the other hand, household solid waste collection does not significantly explain the variance in diarrhoeal disease incidence. This is intuitive as the environmental and sanitary conditions in urban slums tend not to vary much at the household level [49]. Community-led sanitation interventions have a significant effect on the incidence of diarrhoea [50, 51].

Contrary to our expectations, households where children are present have a lower likelihood of reporting diarrhoea. The same observation is made when presence of infants instead of children in a household is considered in the model; albeit, the existing literature report higher diarrhoea morbidity and mortality among children [6, 20, 36, 52]. It is possible that households with infants and children may adopt more hygienic practices to protect them from diarrhoeal diseases in urban poor contexts, with plausible explanation of their exposure to the WASH practices at school. This, however, requires further studies on age-specific diarrhoeal disease epidemiology in poor urban settings.

These findings provide some insights into food types and sources, and household diarrhoeal incidence in an urban poor setting in Ghana. Limitations to the study include having only household-level, rather than individual-level data on diarrhoea incidence. Thus, we had to link each individual's dietary intake to their household's experience with diarrhoea. This resulted in non-household heads being removed from the study. However, this approach added richness to our understanding of the topic since household heads tend to consume the same foods as their entire household. Another is the reliance on self-reported hygiene behaviours such as handwashing. Further research can include observational data on these variables. This study has shown that out-of-home foods can increase the risk of diarrhoea, hence, campaigns/programmes to promote hygienic practices when cooking certain foods must be considered.

## Conclusion

This study examined the influence of household source of food on the incidence of diarrhoea. The results reveal that the type and source of food significantly influences the incidence of diarrhoea in households. Consuming out-of-home cooked rice-based staples, in particular, is associated with a higher likelihood of diarrhoeal disease. In addition, eating staple balls such as fufu, banku and kenkey, both at home and outside has a likelihood of transmitting diarrhoea which highlights the need for hygienic practices during preparation, storage and consumption in all settings. Aside from the sources of food, the source of household water and the presence of children under five years significantly predict the incidence of diarrhoea in households in poor urban communities. The findings from this study have implications for expanding food safety interventions beyond street foods to home-cooked foods in urban poor communities.

## Supporting information

**S1 File.**
(ZIP)

## Acknowledgments

The authors are grateful to Prof. Ama de-Graft Aikins for her insights on the manuscript, particularly during the conceptualization stage. They also wish to thank the field assistants for

data collection efforts and all community members in the three study sites for their responsiveness during data collection.

## Author Contributions

**Conceptualization:** Reuben Tete Larbi, D. Yaw Atiglo, Maame B. Peterson, Adriana A. E. Biney, Naa Dodua Dodoo.

**Data curation:** Adriana A. E. Biney, Naa Dodua Dodoo, F. Nii-Amoo Dodoo.

**Formal analysis:** Reuben Tete Larbi, D. Yaw Atiglo, Adriana A. E. Biney.

**Funding acquisition:** F. Nii-Amoo Dodoo.

**Investigation:** F. Nii-Amoo Dodoo.

**Methodology:** Reuben Tete Larbi, D. Yaw Atiglo, Maame B. Peterson, Adriana A. E. Biney, Naa Dodua Dodoo, F. Nii-Amoo Dodoo.

**Project administration:** F. Nii-Amoo Dodoo.

**Supervision:** Adriana A. E. Biney, Naa Dodua Dodoo, F. Nii-Amoo Dodoo.

**Writing – original draft:** Reuben Tete Larbi, D. Yaw Atiglo, Maame B. Peterson, Adriana A. E. Biney, Naa Dodua Dodoo.

**Writing – review & editing:** Reuben Tete Larbi, D. Yaw Atiglo, Maame B. Peterson, Adriana A. E. Biney, Naa Dodua Dodoo, F. Nii-Amoo Dodoo.

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
