## [Decision Letter · Decision Letter 0]

16 Oct 2020

PONE-D-20-09644

Household food sources and diarrhoea incidence in poor urban communities, Accra Ghana

PLOS ONE

Dear Dr. Biney,

Thank you for submitting your manuscript to PLOS ONE. After careful consideration, we feel that it has merit but does not fully meet PLOS ONE’s publication criteria as it currently stands. Therefore, we invite you to submit a revised version of the manuscript that addresses the points raised during the review process.Please submit your revised manuscript by Nov 30 2020 11:59PM. If you will need more time than this to complete your revisions, please reply to this message or contact the journal office at plosone@plos.org. Please include the following items when submitting your revised manuscript:

We look forward to receiving your revised manuscript.

Kind regards,

Khin Thet Wai, MBBS, MPH, MA (Population & Family Planning Resear

Academic Editor

PLOS ONE

Journal Requirements:

3.We note that you have indicated that data from this study are available upon request. PLOS only allows data to be available upon request if there are legal or ethical restrictions on sharing data publicly. For more information on unacceptable data access restrictions, please see http://journals.plos.org/plosone/s/data-availability#loc-unacceptable-data-access-restrictions.

Additional Editor Comments (if provided):

Minor grammatical errors throughout the manuscript need correction.

In Table 3, please move the values of Model Chi square and Nagalkerke Chi Square results at the header to the footnote.

Reviewers' comments:

Reviewer's Responses to Questions

**Comments to the Author**

1. Is the manuscript technically sound, and do the data support the conclusions?

Reviewer #1: Yes

Reviewer #2: Yes

2. Has the statistical analysis been performed appropriately and rigorously? 

Reviewer #1: Yes

Reviewer #2: Yes

3. Have the authors made all data underlying the findings in their manuscript fully available?

Reviewer #1: Yes

Reviewer #2: Yes

4. Is the manuscript presented in an intelligible fashion and written in standard English?

Reviewer #1: Yes

Reviewer #2: Yes

5. Review Comments to the Author

Reviewer #1: Introduction - It would be helpful to have some statistics on the prevalence of diarrhoea in Ghana specifically from national health surveys, to compliment the studies referenced in line 69 – 80

Line – 97: should be “in public places” not “at”

Line 98: again should read “in public places” not “at”

Line 120: “lay hands on these fruits” – consider being concise and rephrase as “purchase”

Line 129: add “acceptable/recommend” before standards

Line 131 to 133 - “In addition, some street foods are more prone to 132 microbial contamination than others due to the preparation and handling process, sources of raw materials and the mode of serving the prepared food” is repetition of sources of contamination listed in lines 99 – 102

Line 148: “without necessarily through food” - incorrect grammar

Line 152: flies (including cockroaches) – wording suggests cockroaches are a type of fly – consider rephrasing to “insects (flies and cockroaches)”

Methods: In the results, diarrhoea incidence is reported at household level, but in the methods (lines 216 – 221) it is not explained that a household member having diarrhoea was recorded as case of diarrhoea in that home – in other words the methods make it seem like diarrhoea was coded at individual level when it seems to have been coded at household level

Results: Please explain how the survey categorized wealth into the 3 categories

Reviewer #2: Household food sources and diarrhoea incidence in poor urban communities, Accra Ghana

This is an interesting study that identified the sources and categories of food that have the inclination to transmit diarrhoeal disease pathogens. The study classified common tropical foods by source and frequency 86 of consumption vis-a-vis the likelihood of contracting diarrhoeal diseases.

I have the following comments to improve the work

Abstract

1.L32 ……. that have socio-economic characteristics mimicking typical low income communities in the Global South,,,,,how was this done? Did you first carry out a survey in the study area to select such households before the actual study?

2.L173… why 3?

3.L173…were there only 3 densely populated poor urban communities in Accra?

4.L173 …how were the communities selected?

5.L179-184 ….provide citation

6.L191 how was the random selection done?

7.L192…what was the basis for the numbers of EAs in each communities

8.L216 Provide a citation for the definition except it is an operational definition

9.L253…what informs the exact value of k between 0.01 and 99.9

10.L275….. The univariate analysis indicates 506 households included in the study. This is contrary to Line 196-7 where you stated “The total number of household and individual interviews conducted was 660 and 782, respectively.”….Harmonize, perhaps L196-7 was “identified for interview” and not interviewed

11.L 209…Provide the ethical clearance number

12.Table 1: Solid waste disposal and Fruits did not add to 100%

13.Table 2:

•It is non-informative to include both yes and no columns, delete the “No” column

•Remove the all (100.0) in column 4

14.Table 3: there are possibilities of multicollinearity in the explanatory variables used in the multiple regression. How did you control for such?

•Did authors assess the relationship between individual explanatory variables before choosing them as candidate variables in the multiple regression model?

•It is usual to first do this assessment (bivariate regression) and include only the variables that were significant at a specified level (say 10 or 20%) before you run the multiple regression. The outputs of the bivariate regression are the odds ratio, sometimes called crude odds ratio (COR) while those from the multiple regression are called adjusted odds ratio (AOR)

•Authors should specify the ones presented in Table 3.

6. PLOS authors have the option to publish the peer review history of their article (what does this mean?). If published, this will include your full peer review and any attached files.

Reviewer #1: No

Reviewer #2: No

---

## [Author Response · Author response to Decision Letter 0]

14 Dec 2020

Dear Academic Editor:

My co-authors and I are grateful for your decision for us to resubmit the manuscript titled ‘Household food sources and diarrhoea incidence in poor urban communities, Accra Ghana’ for consideration for publication in your journal. 

We have addressed the reviewers’ comments which are indicated below. Once again, we are grateful for you considering our manuscript for publication and look forward to a positive response.

Yours sincerely,

Adriana Biney (Corresponding author)

Response to Reviewers

• Reviewer #1: Introduction - It would be helpful to have some statistics on the prevalence of diarrhoea in Ghana specifically from national health surveys, to compliment the studies referenced in Lines 69 – 80

Response: Statistics on the prevalence of diarrhoea in Ghana, specifically from the 2014 Ghana Demographic and Health Survey, have been provided in Lines 50-52

Line – 97: should be “in public places” not “at”

Response: This has been corrected

Line 98: again should read “in public places” not “at”

Response: This has been corrected

Line 120: “lay hands on these fruits” – consider being concise and rephrase as “purchase”

Response: This has been corrected; “lay hands on these fruits” has been replaced with “purchase”

Line 129: add “acceptable/recommend” before standards

Response: “recommended” has been inserted

Line 131 to 133 - “In addition, some street foods are more prone to 132 microbial contamination than others due to the preparation and handling process, sources of raw materials and the mode of serving the prepared food” is repetition of sources of contamination listed in Lines 99 – 102

Response: This has been amended to read “contamination pathways in the preparation and serving of the types of” in Line 137

Line 148: “without necessarily through food” - incorrect grammar

Response: This has been rephrased to read “directly and not necessarily through food”

Line 152: flies (including cockroaches) – wording suggests cockroaches are a type of fly – consider rephrasing to “insects (flies and cockroaches)”

Response: The sentence has been rephrased to read “flies (and other insects such as cockroaches)”

Methods: In the results, diarrhoea incidence is reported at household level, but in the methods (lines 216 – 221) it is not explained that a household member having diarrhoea was recorded as case of diarrhoea in that home – in other words the methods make it seem like diarrhoea was coded at individual level when it seems to have been coded at household level

Response: The unit of analysis is the household. Any individual that had diarrhoea was recorded as diarrhoea incidence for that household. This has been clarified in Lines 229-234. 

Results: Please explain how the survey categorized wealth into the 3 categories

Response: Household wealth was computed using assets owned by the household such as car, mobile phone, radio, television and fishing boat. We also included the materials used to construct the dwelling in the measurement of wealth. A household wealth index was created using the first factor generated from a principal component analysis. The index was then grouped into terciles; the lowest third in the index were labeled as poor households, the moderate households were the middle third, and the top third in the index being the rich. The revised manuscript has been updated from Lines 278-284

• Reviewer #2: Household food sources and diarrhoea incidence in poor urban communities, Accra Ghana

This is an interesting study that identified the sources and categories of food that have the inclination to transmit diarrhoeal disease pathogens. The study classified common tropical foods by source and frequency 86 of consumption vis-a-vis the likelihood of contracting diarrhoeal diseases.

I have the following comments to improve the work

Abstract

1.L32 ……. that have socio-economic characteristics mimicking typical low income communities in the Global South,,,,,how was this done? Did you first carry out a survey in the study area to select such households before the actual study?

Response: We reviewed the literature on the socioeconomic and demographic characteristics of the study sites as detailed in Lines 182-195 and compared with literature of low-income settlements in the global south (such as UN-Habitat World Cities Report, 2016). This showed that low income communities in the Global South have limited household toilet facilities, water access, poor waste management and have widespread informal food market. 

2.L173… why 3?

Response: The three communities (James Town, Ussher Town and Agbogbloshie) were chosen for the field site so that the research team would be able to compare the characteristics and effects of urban poverty between communities of different migratory and natal circumstances. In other world regions where field research has been carried out in urban poor settings (for example, Kenya and Brazil), the settings are often homogenously slum or non-slum. In our setting, James Town and Ussher Town, spatially close to each other, constitute the Ga-Mashie Traditional Area and are largely inhabited by indigenous Ga people who have resided for generations in those locales whereas Agbogbloshie is typically a migrant slum (though a small fraction of residents trace their roots to as far back as the 1960s). The geographical proximity of the two indigenous communities and Agbogbloshie allowed the study of urban poverty in the two very different settings of an indigenous community setting with generations of inhabitants and a more recent settlement with many migrants, but both with similar levels of poverty. Though other communities also met these criteria, the main reason for the choice of these in particular was geographical proximity to each other. This has been clarified in Lines 179- 182 in the revised manuscript.

3.L173…were there only 3 densely populated poor urban communities in Accra?

Response: The rationale and selected criteria have been detailed in the previous comment, please. This has been clarified in Lines 179- 182 in the revised manuscript.

4.L173 …how were the communities selected?

Response: The rationale and selected criteria have been detailed in the previous comment, please. This has been clarified in Lines 179- 182 in the revised manuscript.

5.L179-184 ….provide citation

Response: Citation have been provided in Lines 190 and 192

6.L191 how was the random selection done?

Response: Five, eight and 16 enumeration areas (EAs) in Agbogbloshie, James Town and Ussher Town, respectively, were randomly sampled proportionate to the number of EAs based on the population sizes in the localities. The study sites were randomly using the random number generator.

7.L192…what was the basis for the numbers of EAs in each communities

Response: We used EAs demarcated by the Ghana Statistical Service for national sample surveys and censuses. This has been clarified in the revised manuscript Lines 201-202

8.L216 Provide a citation for the definition except it is an operational definition

Response: This is an operational definition. 

9.L253…what informs the exact value of k between 0.01 and 99.9

Response: Households with scores of ‘0’ ate the staple foods only outside the home; those with a score of 100 ate the staple foods exclusively at home, and remaining households between these extremes (i.e. 0.1 and 99.9) ate in both settings to some degree. This has been clarified in Lines 265, 268 and 269

10.L275….. The univariate analysis indicates 506 households included in the study. This is contrary to Line 196-7 where you stated “The total number of household and individual interviews conducted was 660 and 782, respectively.”….Harmonize, perhaps L196-7 was “identified for interview” and not interviewed

Response: The total number of household and individual interviews conducted was 660 and 782, respectively. For this paper however, 506 households were used; households that did not have individuals responding to the food consumption schedule were excluded. This has been clarified in Lines 208-211 in the manuscript

11.L 209…Provide the ethical clearance number

Response: ethical clearance number (105/12-13) as been inserted in Line 224

12.Table 1: Solid waste disposal and Fruits did not add to 100%

Response: This has been corrected (Table 1). An “other” category which was omitted has been inserted.

13.Table 2:

•It is non-informative to include both yes and no columns, delete the “No” column

Response: The “No” column in Table 2 has been deleted

•Remove the all (100.0) in column 4

Response: All the (100.0) in column 4 have been deleted 

14.Table 3: there are possibilities of multicollinearity in the explanatory variables used in the multiple regression. How did you control for such?

Response: The authors ran variance inflation factor tests to assess potential multicollinearity in the explanatory variables. We found the highest VIF value to be 1.39 which falls below the acceptable threshold value of 5. Hence, no variables were collinear and could be used in the regression model. A description of this has been included in the Methodology section, Statistical Analysis sub-section, from lines 290 to 293 and Lines 356- 258 in the results section. 

•Did authors assess the relationship between individual explanatory variables before choosing them as candidate variables in the multiple regression model?

•It is usual to first do this assessment (bivariate regression) and include only the variables that were significant at a specified level (say 10 or 20%) before you run the multiple regression. The outputs of the bivariate regression are the odds ratio, sometimes called crude odds ratio (COR) while those from the multiple regression are called adjusted odds ratio (AOR)

Response: We did not assess the bivariate relationships using crude/unadjusted regression models but rather chose to assess the bivariate results using crosstabulations. We desired to identify the proportions in the various categories and test their significance using the chi-square test. We have since run unadjusted/crude binary logistic regression models and found similar relationships and have chosen to keep the crosstabulation results due to a preference for reporting bivariate results in this manner. 

In addition, although a total of eight independent and control variables were not significantly associated with diarrheal incidence at the bivariate level, we chose to include them in the multivariate regression models. The independent variables - stable balls and fruits, had to be included to assess their relationships with the dependent variable, diarrheal incidence. In addition, sex of household head, educational attainment of household head, study locality, toilet facility, household wealth and handwashing with soap before eating are all essential variables that are theoretically linked to diarrheal incidence in a household. We felt that these must still be included in the final models to assess their effects, if any, on the dependent variable. Staple balls, an independent variable, eventually became significant at the multivariate level. 

A description indicating our reasons for including all explanatory variables has been included in the Methods section, Statistical Analysis sub-section from lines 291-295.

•Authors should specify the ones presented in Table 3.

Response: In Table 3, we present the adjusted odds ratios and not the crude/unadjusted odds ratios. A comment indicating this has been included on line 257-258.

---

## [Decision Letter · Decision Letter 1]

4 Jan 2021

Household food sources and diarrhoea incidence in poor urban communities, Accra Ghana

PONE-D-20-09644R1

Dear Dr. Biney,

We’re pleased to inform you that your manuscript has been judged scientifically suitable for publication and will be formally accepted for publication once it meets all outstanding technical requirements.

Kind regards,

Khin Thet Wai, MBBS, MPH, MA (Population & Family Planning Res.)

Academic Editor

PLOS ONE

Additional Editor Comments (optional):

Reviewers' comments:

Reviewer's Responses to Questions

**Comments to the Author**

1. If the authors have adequately addressed your comments raised in a previous round of review and you feel that this manuscript is now acceptable for publication, you may indicate that here to bypass the “Comments to the Author” section, enter your conflict of interest statement in the “Confidential to Editor” section, and submit your "Accept" recommendation.

Reviewer #2: All comments have been addressed

2. Is the manuscript technically sound, and do the data support the conclusions?

Reviewer #2: Yes

3. Has the statistical analysis been performed appropriately and rigorously? 

Reviewer #2: Yes

4. Have the authors made all data underlying the findings in their manuscript fully available?

Reviewer #2: Yes

5. Is the manuscript presented in an intelligible fashion and written in standard English?

Reviewer #2: Yes

6. Review Comments to the Author

Reviewer #2: Am satisfied with the current version. the authors have addressed all the issues raised in the previous version satisfactorily

7. PLOS authors have the option to publish the peer review history of their article (what does this mean?). If published, this will include your full peer review and any attached files.

Reviewer #2: No

---

## [Editor Report · Acceptance letter]

13 Jan 2021

PONE-D-20-09644R1 

Household food sources and diarrhoea incidence in poor urban communities, Accra Ghana 

Dear Dr. Biney:

I'm pleased to inform you that your manuscript has been deemed suitable for publication in PLOS ONE. Congratulations! Your manuscript is now with our production department. 

Kind regards, 

on behalf of

Dr. Khin Thet Wai 

Academic Editor

PLOS ONE